# Towards a Territorially Just Climate Transition—Assessing the Swedish EU Territorial Just Transition Plan Development Process

John Moodie *, Carlos Tapia , Linnea Löfving, Nora Sánchez Gassen and Elin Cedergren

Nordregio, SE-111 86 Stockholm, Sweden; carlos.tapia@nordregio.org (C.T.); linnea.lofving@nordregio.org (L.L.); nora.sanchezgassen@nordregio.org (N.S.G.); elin.cedergren@nordregio.org (E.C.)
* Correspondence: john.moodie@nordregio.org

**Abstract:** The move towards a climate neutral economy and society requires policymakers and practitioners to carefully consider the core technical, social, and spatial dimensions of a just transition. This paper closely examines the processes undertaken during the development of EU Territorial Just Transition Plans (TJTPs) for the three Swedish regions of Gotland, Norrbotten, and Västra Götaland. The aim is to establish whether the content and actions outlined in the TJTPs were driven by the technical, social, or spatial dimensions of a just transition. The analysis is primarily based on a socio-economic and governance impact assessment conducted in each region as part of the TJTP formulation process. These data are also supported by observations of the TJTP development process by the article authors who were part of the team put together by DG Reform to work with the preparation of the TJTPs. The paper finds that the TJTPs development process was largely driven by technical considerations, rather than spatial and socio-economic issues. This indicates that a more open and inclusive place-based territorial approach to climate transition policy formulation and implementation is required. A balance between the technical, social, and spatial elements of a just transition is needed if policies are going to meet the requirements of local and regional citizens and provide sustainable socio-economic growth and environmental protection, without risks of delocalizing energy-intensive processes to other regions.

**Keywords:** climate transitions; climate justice; regional development; territorial plans; place-based policy

## 1. Introduction

Climate transitions will challenge our economies and societies during the years to come. These transformations provide a great opportunity to avert climate risks and define long-term sustainable socio-economic and environmental development pathways; however, before the benefits of systemic climate transition can be fully realized, societies will have to deal with the short-term socio-economic and governance-related impacts caused by these changes. The impacts of climate transformations will not be neutral from a territorial perspective. On the contrary, due to agglomeration economies and regional specialization trajectories over the past decades, the impacts of climate transitions will be highly concentrated in specific countries and regional areas across Europe and the world [1].

The concept of the 'just transition' is starting to gain momentum as it lays at the heart of the European Union's (EU) Green Deal [2] and the United Nations (UN) Agenda 2030 and Sustainable Development Goals [3]. While the just transition is an amorphous concept defined and interpreted differently by diverse actors, an overview of the literature on this topic highlights three core conceptual dimensions. Firstly, the technical dimension relates to the shift towards climate neutral carbon free technologies; secondly, the social justice dimension focuses on citizen involvement in the transition process, preserving jobs and

protecting the most vulnerable in society from the potentially damaging socio-economic impacts of climate policies; and thirdly, the spatial dimension aims to ensure that transition policies are based on territorial specificities that meet the needs of local and regional citizens [4].

The EU fully recognizes that all three dimensions need to be assessed in relation to the development and implementation of just transition climate processes and policies [1]. The EU's Just Transition Mechanism, under the European Green Deal, is committed to identifying and supporting Europe's most affected regions to cope with the technical, social, and territorial specific impacts of the transition to a low carbon economy, focusing on those regions and sectors highly dependent on fossil fuels and energy-intensive processes. A core instrument of the Just Transition Mechanism is the Just Transition Fund, which aims to facilitate socio-economic diversification in regions most affected by climate transition. EU member state access to the Just Transition Fund, implemented within the remit of EU Cohesion Policy, is conditional on the development of Territorial Just Transition Plans (TJTPs) [1].

TJTPs outline regional climate transition processes up until 2030. They identify which EU member state territories should be supported, the main climate-related challenges they face, and the targeted socio-economic and environmental actions and governance mechanisms needed to help meet the threats and opportunities posed by the transition. Central for the European Commission is that the TJTPs are developed and implemented through open and inclusive processes, based on local and regional knowledge and expertise to ensure that climate transition policies meet the needs of citizens and leave no one behind [1].

In Sweden, the draft TJTPs have been developed in close collaboration between national government representatives, national agencies, regional and municipal authorities, and key sectoral actors. The European Commission and Swedish government identified four counties eligible for the Just Transition Fund based on their dependence on carbon intensive industries, namely, Gotland county, Norrbotten county, Västerbotten county, and Västra Götaland county [5,6]. Based on the EU Commission's and the Swedish Government's assessment, the implementation of the fund in Sweden is to be concentrated on those regions where the most greenhouse emissions-heavy industries are located [7]. The TJTP development process took place between October 2020 and January 2021 and the regional TJTP drafts developed are now in the process of being reviewed and validated by the Swedish government and EU.

Nordregio's research team was tasked by DG Reform to directly assist and support national and regional actors in the preparation of the Swedish TJTPs. As part of this process, Nordregio conducted a territorial socio-economic and governance impact analysis within three of the four eligible Swedish regions, Gotland, Norrbotten, and Västra Götaland. The Swedish Agency for Economic and Regional Growth (Tillväxtverket) conducted the analysis for the fourth eligible region, Västerbotten, and the authors of this paper were not involved in this process. Therefore, Västerbotten is not part of this article.

The impact analysis of the three Swedish regions was based on a mixed quantitative and qualitative research approach, including: (1) a socio-economic impact analysis using a combination of socio-demographic and regional economic assessment methods, including a classic economic base and input-output analyses; (2) a desk-based review of primary national and regional climate strategy documents and transition roadmaps developed by regional industries; and (3) semi-structured interviews conducted with national and regional policymakers, public officials, and key sectoral actors. The empirical results presented in Section 6 are based on these analyses.

This paper aims to identify the main technical, social, and spatial challenges each region faces in relation to the green transition and establishing which of the core just transition dimensions informed the overall direction of the contents of the TJTPs. The analysis is primarily based on the socio-economic and governance impact assessment conducted in each region as part of the TJTP formulation process. This information is

supported by direct observations made by the authors, who were directly involved in the TJTP formulation process.

The paper is structured as follows. The first section examines the evolution and key features of the just transition concept, closely examining their central technical, social, and spatial dimensions. This is followed by a discussion of how these different dimensions are interpreted and reflected on within key EU and Swedish climate transition policy documents. The next section provides an empirical overview of the territorial analysis conducted to inform the development of TJTPs for the three Swedish regions of Gotland, Norrbotten, and Västra Götaland, focusing on the main economic, social, and governance challenges presented by the transition. The discussion section reflects on these empirical findings, particularly to what extent the final TJTPs were driven by technical, social, and spatial dimensions of a just transition. We conclude by providing some reflections on the future direction of climate transition policies, in particular the need to take an open and inclusive place-based territorial approach to transition policy formulation and implementation, and the importance of policies striking a balance between the technical and spatial and social justice elements embedded in the notion of just transition towards climate neutrality.

## 2. Defining the Just Transition

The origins of the term 'just transition' can be traced back to the United States during the 1970s when the leader of the American Oil, Chemical, and Atomic Workers Union, Tony Mazzacchi, encouraged the labor unions and national government to engage in peacetime planning to support wartime workers at risk of losing their jobs due to disarmament. Later, in the 1980s, Mazzacchi argued that a "super fund for workers" was required to provide financial and education support for blue-collar workers at risk of unemployment in industries threatened by new environmental legislations [8]. In the 1990s, the 'super fund' was retroactively described as a 'just transition fund' and the term was officially endorsed by various North American labor organizations. In the 21st century, the concept of just transition has become synonymous with the global policy discourse around the need to combat climate change and the shift towards a climate resilient low-carbon economy [9].

The term 'just transition' has been defined and applied differently depending on the context in which it is being used in and who is using it. The just transition concept has been used as a framework by the International Trade Union Confederation (ITUC) to strike a balance between meeting climate and environmental objectives, while protecting and equipping workers whose jobs, livelihoods, and communities are most at risk from climate change or climate interventions [10]. This is confirmed by Rosemberg [11] who notes that the just transition "can be understood as the conceptual framework in which the labour movement captures the complexities of the transition towards a low-carbon and climate-resilient economy, highlighting public policy needs and aiming to maximize benefits and minimize hardships for workers and their communities in this transformation".

In a report by the 'Just transition initiative', it is stated that the methods for achieving just transitions are unclear and that the outcome is dependent on how the transition is implemented in terms of scale, context, and time [9]. The report also addresses the importance of understanding the core principles of the just transition, while at the same time accepting the range of definitions held among stakeholders. While different actors define the just transition differently, including international organizations, unions, environmental lobbyists, and civil society groups, there is broad agreement that its implementation depends on more equitable and inclusive governance processes, policies, and investments.

The concept 'just transition' builds on two related concepts, namely social justice and socio-technical transitions. The term 'transition' was applied and developed further by Frank Geels in his 2002 paper on innovation and climate adaptation [12]. In this paper, Geels introduced the notion of 'sociotechnical transitions' and explained why and how these occur. Geels argues that sociotechnical transitions are an outcome of recurrent innovation and technology substitution processes, such as those required to move from a

fossil dependency to climate neutrality. Geels explains that technical transitions occur as the outcome of linkages between developments at multiple levels (multilevel perspective). Here, he distinguishes between 'sociotechnical regimes' and 'sociotechnical landscape', the former being a group of key stakeholders and the latter an asset of heterogeneous exogeneous and endogenous factors at play. These include determinants like oil prices, economic growth, conflicts, migration, broad political coalitions, cultural and normative values, environmental problems, or climate change. According to Geels, radical innovations are developed when ongoing processes at the levels of regime and landscape create a 'window of opportunity' [12]. These windows may be created by tensions in the sociotechnical regimes or by shifts in the landscape which put pressure on the regime. In the case of the just transition, we can interpret climate change as the sociotechnical landscape, which has put pressure on businesses, stakeholders, and institutions (the sociotechnical regimes) to make changes.

The term 'just' in just transitions relates to the concept of social justice developed by philosophers and thinkers, including Rawls, Locke, Rousseau, and Kant [13]. This literature identifies a tension between two important paradigms of social justice, namely the distributional and procedural components of justice. These two paradigms are not considered as opposites but are both required to uphold justice. The distributional paradigm relates to the equal distribution of goods, services, and opportunities, as well as burdens [14], while the procedural paradigm revolves around just institutions and procedures, focusing on the extent to which individual and organizational actors are able to meaningfully participate in the decision making process [15]. Just procedures are necessary, but not sufficient for the fairness of the outcome, while attention to the outcome may mask the injustices of the process. Social justice is, therefore, a normative concept which implies that society and the state should strive towards achieving social justice for all citizens. Recognitional considerations form an important part of the procedural dimension with an emphasis on the role of disadvantaged and vulnerable minority groups whose voices are often left unheard in decision making processes, including policy making [4].

Both the 'socio-economic' and 'just' elements are clearly reflected in the just transition guidelines developed by the ITUC and International Labor Organization (ILO) [16]. These outline the need for high levels of investment in low-carbon technologies; a social dialogue and consultations between policymakers and affected groups; proactive labor market policies based on social protection and worker rights, including financial support and retraining opportunities for the unemployed; early-stage research into the potential socio-economic impacts of climate policies; and the development of local economic diversification plans [10]. The guidelines also make a distinction between social investment policies (e.g., active labor market policies, training and re-skilling policies to increase workers' employability in a greener economy) and social protection policies (e.g., unemployment and minimum income benefits) and mean that both are needed in a just transition framework.

The local and regional implications of the climate transition also highlight the need for a spatial dimension to justice. Spatial justice emphasizes how macro forces can cause local injustices, which resonates with how some regions or municipalities dependent on industries with high emissions will be unevenly impacted by the climate transition. The spatial element of the just transition has been a focal point for international policymakers who have highlighted the importance of regional context and place-based responses in the development and implementation of spatially just transition plans. The different dimensions of the just transition in relation to EU policy are discussed more in the following section.

## 3. The EU Just Transition—A Territorial Specific Approach

The EU has a long history of supporting and guiding regions in industrial transition. The original European Coal and Steel Community provided technical and economic assistance for different regions to undergo transitions or reconstructions of industries. More recently, the 2017 Initiative for Coal Regions in Transition focused on the transition of European coal regions, which included the development of the EU Just Transition Platform

to facilitate the exchange of best practices and discuss strategies and projects with the potential to kick-start the transition process in declining coal regions [17].

In 2019, the European Commission adopted the European Green Deal [2]. The strategy states that the European Union should have no net emissions of greenhouse gases in 2050 to reach the Paris Agreement's goal of holding warming "well below" 2 °C and pursuing efforts to keep warming below 1.5 °C. To achieve this goal and ensure that 'no person or place is left behind' as a consequence of climate transitions, the Just Transition Mechanism (JTM) was introduced [18]. Its main objective is to ensure that the transition takes place in an effective and fair manner. The JTM is split into three main pillars. Pillar one, the Just Transition Fund, provides financial support to those regions and sectors most dependent on fossil fuels, and thereby also more affected by the transition. Pillar two, the InvestEU Just Transition Scheme, provides budgetary support for private sector investments that support transition. Finally, pillar three, the Public Loan Facility, supports public sector transition investments.

The Commission notes that support will be available to all member states, focused on regions that are the most carbon-intensive and people and citizens most vulnerable to the transition. The JTM will protect them by:

- supporting the transition to low-carbon and climate-resilient activities;
- creating new jobs in the green economy;
- offering re-skilling opportunities;
- investing in public and sustainable transport;
- providing technical assistance;
- investing in renewable energy sources;
- improving digital connectivity;
- providing affordable loans to local public authorities; and
- improving energy infrastructure, district heating, and transportation networks [1].

The aims and objectives outlined by the JTM indicate an attempt to balance the socio-technical and social justice related elements of the just transition. The socio-technical elements are reflected in the EU's commitment to investing in carbon-neutral and environmentally friendly production technologies, transport, renewable energies, and digitalization, whereas the social elements are highlighted in the worker protection rights and reskilling opportunities offered. In a key passage within a communication document on the EU Green Deal, the Commission further outlines the core social elements of the just transition:

> "This transition must be just and inclusive. It must put people first, and pay attention to the regions, industries and workers who will face the greatest challenges. Since it will bring substantial change, active public participation and confidence in the transition is paramount if policies are to work and be accepted. A new pact is needed to bring together citizens in all their diversity, with national, regional, local authorities, civil society and industry working closely with the EU's institutions and consultative bodies." [2]

In this excerpt, there is a strong emphasis on the social justice element of the transition, particularly in relation to the need for open and inclusive social dialogue with citizens to promote the trust and acceptance of policies.

Sabato and Fronteddu [19] argue that the understanding of the just transition in the European Green Deal promotes social investment policies (e.g., active labor market policies, training and re-skilling policies to increase workers' employability in a greener economy) in favor of the need to ensure the protection of citizens through traditional social protection policies (e.g., unemployment and minimum income benefits). [1] They argue that EU's policy approach to the transition is not consistent with the Guidelines of the ILO [16] which emphasize the need to place territorial/sectorial policies and social investment policies within a strong social protection system guaranteeing social rights to all citizens. As will be discussed further, the Swedish welfare system is relatively strong in comparison to many

other countries which means that social investment policies to some extent will be placed in a social protection system, as guided by the ILO. However, this will not be the case for all member states. That said, the Communication of the European Green Deal refers to the European Pillar of Social Rights principles as a reference framework to promote equal opportunities and access to the labor market; fair working conditions; and social protection and inclusion [20], which emphasize a broader objective of the promotion of social rights, even though the practical implications of this are unclear.

Other priorities are implicit and embedded in the concept of just transition, as formulated in the JTM. The spatial and territorial dimensions of the just transition is one of those embedded principles. This priority is regularly advocated within the EU transition policy documents, with the Commission highlighting the important role of regions and cities in guiding just transition processes: "Citizens, depending on their social and geographic circumstances, will be affected in different ways. Not all Member States, regions and cities start the transition from the same point or have the same capacity to respond. These challenges require a strong policy response at all levels." [2]. The territorial focus of the EU's just transition plans is in keeping with the Commission's recent emphasis on territorial governance and place-based policymaking, such as the development of regional smart specialization strategies, in addition to applying the concept of 'active subsidiarity' which advocates a central role of regions and cities in EU policy formulation and implementation [21].

The region-specific focus also links climate transition to the concept of 'regional resilience' of local communities and regions [22]. Regional resilience refers to a set of regional and local economic, social, and institutional traits that characterize the ability of regions to respond to a shock and maintain system stability and durability, as well as adapt to structural changes and move to new development pathways [23]. Although the carbon transition is often closely linked to more or less 'spontaneous' technological changes and innovations, it can be considered a policy-induced shock to regions, with new rules putting pressure on policymakers, industries, businesses, workers, and citizens alike (e.g., new tax regimes, shift on investments, trade deals, revised regulations and laws, or as in the case of the JTM, supportive investments and incentives). Many variables are identified as important to endure an economic shock, such as the existing economic path of a region, regional economic structures, resources, capabilities, and competences. It can also be business cultures and any supportive measures implemented by different institutions at national and subnational levels (e.g., welfare policies and programs). The OECD has identified four areas that drive regional resilience, including clear leadership and management; strategic and integrated approaches; public sector skills; and open and transparent governments [24]. This highlights the importance of regional governance processes and actors in formulating and implementing a just transition.

The significant role of regional structures and actors is most visible in the Commission proposals for member states to develop TJTPs in regions most affected by climate issues. The TJTPs are to be developed through a dialogue between the European Commission, national government representatives, and regional actors. As the European Commission notes, "these plans set out the challenges in each territory, as well as the development needs and objectives to be met by 2030" [1]. They identify the types of operations envisaged and specify governance mechanisms. The approval of the TJTPs opens the doors to dedicated financing under the pillars of the JTM. EU member states are currently in the process of developing and ratifying their TJTP draft proposals. The following sections examine and analyze the process of developing TJTPs in three Swedish regions, focusing on the extent to which the TJTPs developed were driven by technical, social, and territorial place-based dimensions which meet the needs and requirements of local stakeholders and citizens.

## 4. Research Methods

The following analysis of the processes undertaken to develop TJTPs in three Swedish regions is primarily based on an economic, social, and governance impact assessment conducted by Nordregio researchers at the request of DG Reform and the Swedish government. This assessment was conducted using a combination of qualitative and quantitative research methods. The three analytical strands presented below were performed on each of the three regions in scope.

A socio-economic analysis included a regional economic base analysis and an impact assessment. The economic base analysis was conducted to characterize regional economies and their evolution prior to the adoption of the TJTP. This included the calculation of location quotients for selected sectors and a shift-share analysis focusing on employment patterns in the industries under investigation. Such empirical analyses were performed using regional statistics on economic production (in value added) and employment (in full-time equivalents) provided by Statistics Sweden and the regional statistical offices. The analysis of potential socio-economic impacts linked to the decarbonization of the industries under scrutiny were evaluated by means of an input-output analysis. This assessment was performed on the symmetric input-output tables (SIOT) for year 2018 provided by Eurostat. The regional employment and value-added effects were estimated through regionalized input-output coefficients. The potential impacts were modeled under worst-case scenarios that assumed a discontinuation of the activities of the major industrial emitters in each region.

A social analysis looked at demographic dynamics, including aging processes, migration trends, population projections, and statistics related to educational outcomes in the different counties. Particular attention was put on the capacity of regional economies to attract and retain trained workforce. Gender aspects were considered both from the demographic balance as well as from labor segregation perspectives.

A governance assessment was based on a desk-based examination of primary EU, national, and regional level climate strategy documents, including climate roadmaps developed by key sectoral actors within the three Swedish regions. The data and information were supplemented by authors' observations of the TJTP development process. Authors formed part of the team assembled by DG Reform and Swedish government representatives to produce first drafts of the TJTPs. As part of the process, the Nordregio research team had first-hand access and involvement in meetings in which they could observe discussions between key stakeholders including government representatives, national agencies, and regional and local public authorities. Finally, semi-structured interviews with national and regional level actors were also performed which allowed for more detailed discussion in relation to the main economic-, social-, and governance-related issues posed by climate transition policies.

## 5. The Swedish Climate Policy Framework

In 2017, the Swedish Parliament adopted a climate policy framework outlining Sweden's approach for complying with the Paris Agreement. The framework sets ambitious targets for climate and energy and goes further than the EU's 2050 climate neutrality foci and current energy and climate objectives for 2030. Sweden has committed to reducing all net emissions of greenhouse gas (GHG) into the atmosphere to zero by 2045 and using 100 percent renewable energy in 2040. The Climate Policy Framework also includes a Climate Act which regulates the government's climate policy work, including its overall aims and how they should be implemented. The Framework is based on the development of a climate policy action plan every four years [5]. This plan should demonstrate how the government's overall policies in all relevant spending areas contribute to achieving the 2030 and 2040 milestones and the long-term emissions target by 2045. A Climate Policy Council has also been established as an authority in the form of an independent interdisciplinary expert body that is tasked with evaluating how the government's overall policy is compatible with the climate objectives decided by the parliament and government.

The term 'just transition' is rarely used within Swedish national and regional climate and energy strategies, except for documents connected to the European Just Transition Mechanism. The terms 'climate transition' and 'green transition' are instead commonly used, often linked to sustainable development and the UN's Agenda 2030. There are very few public policies and policy documents exploring the socio-economic impacts of the green transition in Sweden. However, many of the goals and objectives outlined within the climate framework and other initiatives focus on the technical elements required to support the transition. This includes financial support to improve energy efficiency, financial support for industrial companies to reduce their emissions through technical advances, a digitalization strategy, and a strategy for a circular economy. The Swedish Trade Union Confederation (LO) has argued that the just transition in Sweden has mainly focused on technological questions, and it is important that transition plans reflect the needs of both industry and workers affected by climate policies [25].

In response, the undersecretary to the Minster for Climate and Environment in Sweden highlighted in 2020 that the ambitious Swedish climate policies need to develop in parallel with social justice, social security, equity, and gender equality [25]. In June 2020, the Ministry for Foreign Affairs also published a report that operationalized the "leave no one behind" principle from the UNs Agenda 2030 in Sweden. The report stresses seven main messages: realizing human rights and gender equality; strengthening empowerment and participation; advancing the transition towards resource-efficient, resilient, and climate-neutral economies; promoting multidimensional poverty reduction; promoting social dialogue and decent work; progressively realizing universal social protection; and improving data and monitoring. The report states that "special attention must be paid to the social and gender dimension of the transition, in order to ensure that no one, particularly people living in poverty, is left behind when society implements measures to become climate-neutral" [26].

The work to reach the environmental objectives has been designed as a concerted effort across the whole of society, including public agencies, business communities, stakeholder organizations, and, not least, individual citizens. Many regions, County Administrative Boards, and municipalities have also developed climate and energy strategies [27–29]. As part of Fossil-Free Sweden, 22 sectors have developed their own road maps for fossil-free competitiveness, including the sectors in focus for the JTF, the cement industry, the mining and minerals industry, the steel industry, and the petroleum industry. Regional and local authorities play a central role in the implementation of climate and energy transition plans in Sweden [30]. The implementation of the objectives outlined in the Swedish Climate Framework draws on multi-level governance processes with a central role for regional public authorities and industries with territorial knowledge and expertise. Table 1 provides an overview of the key administrative levels involved in the development and realization of the TJTPs in Sweden, along with their formal tasks.

**Table 1.** Overview of the key administrative levels in the process outlining the TJTPs.

| **European** | European Commission<br>**Support for the development of the TJTPs is provided by DG REFORM.** | | |
|---|---|---|---|
| **National** | **National authorities**<br>The national authorities are responsible for implementing the decisions made by the Parliament and the Government.<br><br>**Role in TJTP process:**<br>An authority group has been established to lead and coordinate the process, align priorities as well as to assist with expertise and data. These consist of:<br>• Swedish Agency for Economic and Regional Growth (managing authority)<br>• Swedish Public Employment Service<br>• Swedish Energy Agency<br>• Swedish Environmental Protection Agency | | |
| | **County Administrative Boards**<br>National authorities operating at county level, responsible for the state administration in the county (in those areas where no other authority is responsible for special administrative tasks). The County Administrative Boards shall work to ensure that national goals have an impact in the county, while also taking into account regional conditions.<br><br>**Role in TJTP process:** The county administrative boards have the mission to promote, coordinate, and lead the regional work in the implementation of the government's policy regarding energy conversion and reduced climate impact with a long-term perspective. Together with the regions, they have important roles in the implementation of the TJTP. These have participated in meetings at regional level and provided input during the work. | | |
| | County Administrative board of Norrbotten | County Administrative board of Västra Götaland | County Administrative board of Gotland |
| **Regional/ County** | **Regional Authorities**<br>Led by political assemblies with responsibilities for, e.g., health care and social care, public transportation, and regional development. Counties and regions cover the same geographical area.<br><br>**Role in TJTP process:** The regions have the responsibility for regional development and thus a key role in the implementation of the plan. The region has an active role in moving the regional economy from fossil dependence to a sustainable society and an especially important role in linking sectors as well as the regional and national level in the innovation system to create the necessary triple-helix collaborations. | | |
| | Region Norrbotten | Region Västra Götaland | Region Gotland |
| **Local** | **Local authorities**<br>Local government responsible for, e.g., primary and secondary school, preschool activities, elderly care, physical and comprehensive planning, roads, water and sewage issues, and energy issues. They also issue different types of permits, such as building permits.<br><br>**Role in TJTP process:** Municipalities play an important role in Sweden's climate work. Due to the proximity to the citizens, their roles for spatial planning and as large employers are the significant climate actors in the work towards set climate goals. The municipalities drive local development in collaboration with companies, organizations, and residents, and will therefore play a significant role in the implementation of the TJTP. The municipalities participated in the drafting of the TJTPs through a written consultation process on the plans and related outputs. | | |
| | • Gällivare municipality<br>• Öxelösund municipality<br>• Luleå municipality | • Lysekil municipality<br>• Stenugnssund municipality | • Gotland Municipality |

**Source**: Government Offices of Sweden, 2020; TJTP Gotland; TJTP Norrbotten; TJTP Västra Götaland.

## 6. Developing EU Territorial Just Transition Plans in Sweden

The industries and counties that are proposed to be covered by the EU's new climate fund are the steel industry in Norrbotten, the cement industry on Gotland, refineries and the petro-chemical industry in Västra Götaland, and the metal industry in Västerbotten. These counties and industries have been selected as they contribute to the largest shares of carbon emissions in Sweden. In absolute terms, Gotland, Norrbotten, and Västra Götaland are accountable for around 19.8 million metric tons of carbon dioxide equivalents (MMTCDE). This represents a third (34.7 percent) of total fossil-based greenhouse gas (GHG) emissions in Sweden [31]. The largest GHG contributor is Västra Götaland (11.9 MMTCDE; 20.8 percent of total GHG emissions), followed by Norrbotten (5.2 MMTCDE; 9.2 percent of total GHG emissions) and Gotland (2.7 MMTCDE; 4.8 percent of total GHG emissions). In 2018, the two Swedish counties with the highest emission intensities in terms of fossil-based GHG emissions per unit of Gross Regional Domestic Product (GRDP) were Gotland and Norrbotten. Västra Götaland ranked sixth in the list. Even if Gotland in economic and employment terms represents a small share of Sweden's economy (0.5 percent and 0.6 percent, respectively), it contributes to almost 5 percent of total GHG emissions at national level. As a result, Gotland is the most carbon-intensive county in Sweden [31].

The diagnostic work for developing the TJTPs in the four selected Swedish regions is taking place in dialogue with different actors within existing collaborative structures across multiple levels of governance. This work is being conducted in dialogue with different actors within existing collaborative structures across multiple levels of governance. The Swedish Agency for Economic and Regional Growth (Tillväxtverket) is tasked by the government to ensure preparations for the Fund. The government has identified the Swedish Public Employment Service, the Swedish Energy Agency, and the Swedish Environmental Protection Agency as especially involved agencies in the preparations. These government agencies have worked in close collaboration with the County Administrative Boards, the regions, and municipalities in the preparation of the TJTPs [7]. Local public authorities have in turn engaged in dialogue with the public and private sector, academia, and other interested parties. The plans are also drafted in dialogue with key industry stakeholders such as Cementa AB and the HeidelbergCement Group representing the cement industry on Gotland; SSAB, Lulekraft AB, and LKAB Kiruna from the HYBRIT initiative in the Norrbotten county; as well as the chemical and refining industries, including Preem AB, Borealis AB, and ST1 in the Västra Götaland county. Other key sectoral agencies, stakeholders, and actors have been an important part of formulating the plans. These include Vattenfall and Svenska Kraftnät in the energy sector, higher education institutions, academia, and other interested civil society and labor organizations.

The TJTPs for each transition region have the same structure, including the following seven core elements; one, an overview of the national climate policy framework, highlighting key national climate targets; two, a national level assessment of the territorial impacts of climate change, identifying which regions within the country contribute the most to gas and carbon emissions; three, an assessment of the potential economic, social, and environmental impacts of the transition on the selected region; four, an analysis of the development needs required in each region to meet climate targets; five, an outline of the main type of actions planned to deliver the regional transition; six, performance-related output and results indicators are highlighted for measuring the impact of the proposed actions; and seven, the governance structured and key stakeholders needed to develop and implement the plans are outlined. The proposed regional just transition fund support actions are shown in Table 2. The outlined actions must contribute to a positive development regarding gender equality, integration and diversity, the environment, and the living conditions of young people. These horizontal criteria form the basis for the Just Transition Fund support efforts, as outlined in the plans.

**Table 2.** Proposed actions in the Swedish Territorial Just Transition Plans.

| Type of Action | Gotland | Norrbotten | Västra Götaland |
|---|---|---|---|
| **Investments for the use of clean energy technologies and infrastructure, reduction of greenhouse gas emissions, energy efficiency, and renewable energy.** | Support the cement industry with fuels substitution to waste-based and bio-based fuels as well as new cement grades and materials in the production. Investments in improving infrastructures for a flexible and robust energy system in the island and the connections to the mainland. | Transition to carbon neutral steel production (support to EU ETS steel industry facilities in Norrbotten). | Transition to the production and use of green hydrogen in production processes in the refinery and petro-chemical industry; increasing production capacity for biofuels to reduce $CO_2$ emissions. |
| **Investments in research and innovation and promotion of advanced technology transfer.** | RD&I for developing efficient and commercially available technology for CCS, alternative fuels and electrification, as well as new cement grades and materials in the production | Innovation for the production of innovation-critical raw materials and materials necessary for a transition to a fossil-free society; RD&I for large-scale energy storage and development and implementation of fossil-free technologies and other alternative energy carriers and raw materials. | Support for mapping Västra Götaland's ability to form a hydrogen cluster; RD&I for new raw materials and secondary materials, including waste streams, for biofuel production as well as separation, use and storage of carbon dioxide (CCS/CCU). |
| **Skills upgrading and retraining of employees.** | - | Mapping of the steel industry's skills needs; support for networks and clusters for skills-enhancing initiatives in the steel industry and its value chain, retraining and skills development of existing and new workforce, skills validation measures and strengthen the companies' strategic work with skills issues. | - |
| **Investments to promote the circular economy, including through measures to prevent and reduce waste, resource efficiency, reuse and recycling.** | Minimize waste and increase recycling in the cement production. | Efforts to promote the use of recycled materials as a raw material; support for environmentally friendly production processes and resource efficiency. | Improve the use of resources and recycling in refineries and the chemical industry; re-use of waste oils (refineries) and chemical recycling of plastics (polyethylene) for chemical industry. |

**Source:** Swedish TJTPs for Gotland, Norrbotten, and Västra Götaland.

Draft versions of the TJTPs were sent out between November 2020 (Gotland and Norbotten) and February 2021 (Västra Götaland and Västerbotten) for collection of comments from relevant authorities, companies, organizations, and municipalities. After compiling the comments from the consultation, revision, and completion of the program proposals from the Agency for Regional and Economic Growth and the Ministry of Enterprise and Industry, the final proposals were presented to the Government Offices for validation in March 2021. The following sub-sections provide an overview of the key findings from economic, social, and governance analysis within the TJTPs, focusing on the main commonalities and differences from across the three selected Swedish transition regions, Gotland, Norbotten, and Västra Götaland.

*6.1. Economic Analysis*

Discussions surrounding the development of the regional EU TJTPs revolved largely around the technical dimension of the shift towards climate neutrality. The focus is on the degree of maturity and feasibility of specific carbon-free technologies, vis-a-vis the socio-economic costs of the transformation. The Swedish national government is committed to devoting the EU Just Transition Fund to the development of climate-neutral processes by existing regional fossil-dependent industries. This vision emerges in the proposed actions within each TJTP, which are skewed towards the technical elements of climate transitions (Table 2). Most of the interventions target specific industrial actors and some have been even designed at plant level. These priorities are also reflected in regional energy and climate policy documents. The primary focus on the technical dimension of the transition is driven by the view that EU transition funding should cover the economic costs of technological change. This might derive from the perception that the Swedish welfare state and other funding sources have the capacity to address territorial socio-economic challenges caused by the transition.

Large fossil fuel-based industries make up an important part of the economic profile of the selected Swedish just transition regions. While these primary industries represent a comparatively modest share of employment and gross value added (GVA), their relevance in specific localities and their contribution to overall regional economic growth is substantial. Moreover, the affected industries provide products with high strategic value, including cement, steel, and basic chemicals. Most of these commodities still have limited substitutability by fossil-free alternatives. This explains why national and regional policy-makers are committed to achieving the technical dimension of transition policies. A failure to support these industries in the shift to climate neutral technologies would have major economic consequences at both national and regional levels.

This situation holds particularly in Norrbotten, where the manufacture of basic metals and fabricated metal products sector (grouped in categories 24–25, according to Euro-stat's Statistical classification of economic activities, NACE Rev. 2) is the sixth largest employer [32]. In total, 2610 persons worked in this sector in 2018. Still, since the iron ore used by local steel plants is sourced locally, steel processing is closely connected to local mining. Combined, the mining and steel sectors clearly dominate the regional economy. In 2018, 6900 persons were directly employed in these industries, which corresponds to 10.7 percent of the total number of jobs in Norrbotten. Employment in both sectors is often concentrated in a small number of large mines and plants. In Kiruna, for instance, around 2175 workers are hired by the state-owned company LKAB, a producer of iron ore, corresponding to 17.5 percent of local jobs [33]. In Gällivare, the same company employs 1175 people (13.5 percent of all employees in the municipality) [34]. Between 2010 and 2018, the manufacture of basic metals and fabricated metals sectors in Norrbotten suffered a decline in employment (436 jobs lost) that contrasts with a period of relatively high economic dynamism in the region and in Sweden as a whole. Still, in comparative terms, the sector performed better in Norrbotten than in other Swedish regions. This suggests that this county provides good conditions for this sector that still account for a very relevant share of jobs in some localities. In Luleå, for example, the SSAB EMEA AB steel plant employs 1325 persons, corresponding to 1.2 percent of all jobs in the region [35]. Based on the employment multipliers from our input-output analysis (Table 3), if the activity of this plant was discontinued as a response to stringent climate policies and regulations, a total of 1965 full-time equivalent (FTE) jobs might be lost in Norrbotten. This includes the initial jobs lost in the SSAB EMEA AB plant itself, as well as 640 indirect jobs lost in the region because of backward linkages in the steel value chain.

**Table 3.** Employment indicators and multipliers for selected economic sectors in Gotland, Norrbotten, and Västra Götaland (2018).

| Indicator | Gotland: Other Non-Metallic Mineral Products (CPA_23) | Västra Götaland: Coke and Refined Petroleum Products (CPA_C19), Chemicals and Chemical Products (CPA_C20), and Basic Pharmaceutical Products and Pharmaceutical Preparations (CPA_C21) | Norrbotten: Basic Metals (CPA_C24) and Fabricated Metal Products, Except Machinery and Equipment (CPA_C25) |
|---|---|---|---|
| Initial employment effect | 0.329 | 0.119 | 0.308 |
| First-round employment effect | 0.051 | 0.057 | 0.104 |
| Industrial support employment effect | 0.017 | 0.210 | 0.045 |
| Production-induced employment effect | 0.067 | 0.034 | 0.149 |
| Simple employment multiplier | 0.396 | 0.092 | 0.457 |
| Type 1A employment multiplier | 1.154 | 1.481 | 1.339 |
| Type 1B employment multiplier | **1.204** | **1.769** | **1.483** |

Note: Employment multipliers were calculated by means of input-output analyses on the symmetric input-output tables provided by Eurostat (naio_10_cp1700). National input-output coefficients were calculated by direct allocation of competing imports. Input-output coefficients were regionalized by applying Flegg's location quotient (FLQ), method proposed by Flegg, Webber, and Elliott (1995). Even if several methods can be used to regionalize the input-output coefficients, it is now well established that the FLQ method can give more precise results than alternative approaches, like SLQ or the CILQ [36,37]. Employment multipliers highlighted in bold represent the extra number of persons employed in all industries in the economy for one extra person employed in the industries under investigation. These multipliers hence summarize the aggregated direct and indirect employment effects for every new job created in the relevant sectors, including initial, first round and industrial support induced output effect. Type 1B employment multipliers were hence used to calculate the employment effects described on the text.

In Gotland, the limestone and non-metallic mineral manufacturing industries are small but important contributors to the island's economy. Even if the number of direct jobs provided by these sectors is rather limited (398 workers in 2018), its relevance for Gotland's economy is considerably greater than for other regions in Sweden. In 2018, these sectors represented 3.2 percent of region's total employment [32]. Based on the location quotients, in employment terms, the industry for non-metallic mineral products is in fact the single most overrepresented economic activity in Gotland's economy. Cement and limestone industries are particularly important for rural Gotland, and particularly the northern part of the island, where the main extractive and processing plants are located. The shift-share analysis performed for 2010–2018 data shows that Gotland's economy has comparatively lower competitiveness levels than other Swedish regions. These conditions also affect the non-metallic manufacturing sector. Even if the sector is performing comparatively better in Gotland than its peers at national level, it is still a very traditional industry with a weaker competitive position in comparison to other sectors and high dependence on national economic dynamics (essentially, its level of output is driven by the demand in the building construction sector). Based on the worst-case scenario that all the 230 FTE jobs in the Cementa AB plant at Slite could be lost as a response to climate-driven decisions, by applying the regionalized employment multipliers for 2018 presented in Table 3, we find that the hypothetical number of direct and indirect FTE jobs lost in the island would be around 277.

Västra Götaland is one of the three largest regional economies in Sweden. The region has an expansive economy and a strong and export-oriented industrial sector. The manufacture of refined petroleum and chemical industry, (NACE sectors 19–20), excluding pharma (NACE 21), represents a small share of the economy in the region. In 2018, these sectors employed 5669 persons (1.5 percent of the total workforce in the region) and, together with the pharmaceutical industry, generated 15,272 million SEK in value added (3.2 percent of

region's total GVA). [32] From a territorial perspective, the largest dependence on economic activity in the petrochemical sector concentrates in a restricted number of municipalities in Västra Götaland. Dependence is higher in smaller communities, particularly Lysekil and Stenungsund, where chemical and refining sectors occupy a large proportion of the working population. In the Lysekil municipality, Preem is by far the largest private employer, with 600 people directly engaged by the company [38]. In Stenungsund, it is estimated that the chemical industry cluster provides at least 2350 direct jobs. [39] In the 2010–2018 period, both the regional economy in general and the petrochemical sector showed positive economic trajectories. The refining sector (NACE 19) maintained a stable workforce over the last decade (from 1430 in 2007 to 1611 in 2018). The chemical sector (NACE 20) oscillated between 4500 employees in 2010 and 4058 in 2018 [40]. According to our shift-share analysis, regional competitive factors seem to be driving the overall good employment performance, particularly in the chemical sector. By applying the Type 1B multiplier on employment data for the refining industry provided by the Västra Götaland region [40], we estimated that the direct and indirect employment effects caused by a hypothetical discontinuation of the activities of the local oil refineries might lead to a potential loss of 2850 FTE jobs in the county. This figure represents around 0.6 percent of total employment in Västra Götaland.

### 6.2. Social Analysis

Contrary to our worst-case scenarios, national and regional policymakers and stakeholders interviewed as part of the TJTP development process stressed that no job losses are anticipated or considered likely as a result of the climate policies introduced in any of the Swedish just transition regions [41]. On the contrary, the transition to a sustainable industrial system could even lead to investments and job creation. This growth potential could present challenges caused by existing socio-economic development trends occurring within the three regions.

Regional labor shortages could pose a significant challenge to climate transition processes due to negative demographic development trends. Population aging is creating a shift in the labor force with the large number of people reaching retirement age leading to higher recruitment needs. The population in Norrbotten is declining; the population size on the county level decreased by around three percent between 2000 and 2020 due to negative population growth and the outmigration of young skilled people [42]. Similarly, in Gotland, the working age population fell from 32,700 persons in 2009 to around 32,100 persons in 2019 [43]. Consequently, both Norrbotten and Gotland today have a smaller labor supply from which companies can recruit employees than ten years ago. These regions are reliant on attracting skilled immigrants to overcome this shortfall, so regional authorities have focused on making these areas more attractive places to live by providing affordable quality housing and better transport accessibility links.

In order to reap the opportunities and benefits presented by the climate transition, access to the right competencies and skills is required. Plans to increase energy efficiency, to electrify industries, to increase the use of biogas, and to develop options for carbon capture and storage require skills that are not always readily available. In Norrbotten, a relatively high proportion of young people, particularly boys, leave school without qualifying for higher secondary education (gymnasiebehörighet). In the school year 2018/2019, around 14 percent of pupils belonged to this group. The proportion of non-qualifying pupils was higher in some smaller rural municipalities such as Pajala and Övertorneå than in the regional centers Luleå and Piteå [44]. Low educational attainments are a challenge for young people who may experience difficulties in finding employment, but they are also a challenge for businesses. Already in 2017, one out of four businesses in Norrbotten stated that they experienced challenges in recruiting people with the right competencies [45], and that this was a major obstacle to development and growth. A positive development in this context is that educational attainment levels in Norrbotten have increased during the last decade. In 2019, 31.6 percent of the adult population had attained post-secondary education.

This proportion is lower than for the Swedish population at national level (37.9 percent in 2019). Nonetheless, it is a substantial change in comparison to 2010, when only 27.3 percent of people in Norrbotten had attended post-secondary education or training [43].

Similarly, in Gotland, 25 percent of employers report difficulties in finding staff, particularly with highly specialized competencies, as educational attainment levels of the population are comparatively low [46]. In Sweden as a whole, more than 38 percent of the population over 16 years have taken part in or concluded post-secondary education. In Gotland, it is only 31 percent [43]. These trends are also reflected in Västra Götaland, where several population groups struggle to enter the labor market due to low levels of education, particularly younger people. Almost one out of five students in the county leave comprehensive schools (grundskola) without qualifying for upper secondary education (gymansieskola). This negatively influences their employment prospects and career options [39]. The proportion of students who do not qualify for higher secondary education has fluctuated during the last years in Västra Götaland and in Sweden as a whole. Västra Götaland has closely followed the Swedish trend, with proportions of non-qualifying students increasing especially rapidly between the years 2015/2016 and 2016/2017. As a result of these trends, all Swedish transition regions need to provide local citizens with access to life-long learning opportunities, vocational training, and reskilling. This requires establishing close links between industries and businesses to ensure that student education courses match the needs of employers.

Women are another target group for sectors affected by the climate transition. The labor markets within core transition industries in Gotland, Norrbotten, and Västra Götaland are gender-segregated to a relatively strong degree. For example, in Gotland, around half of all women work in the public sector, in particular in health care and social care, while around 80 percent of men work in the sectors that are important in the climate transition—including the cement and limestone industry, energy, transport, and the building sector [46]. In Gotland, there are strong educational attainment differences, with men less likely to have obtained higher education than women. This is also the case in Västra Götaland and Norrbotten, where the risk of leaving comprehensive schools without moving on to higher education is more pronounced among young men than among young women. Subsequently, attracting more women to work in these traditional sectors could be one solution to potential labor shortages in these regions [39].

One social group that is particularly vulnerable to climate change and the climate transition in Norrbotten is the Sámi people. Climate change, as well as policies and measures to support the climate transition, affect the context in which the Sámi preserve their unique culture and traditional livelihoods. Changes in temperatures and precipitation influence the conditions for reindeer herding, for instance through changes in food supply for reindeers as well as water and flooding conditions. At the same time, laws and policies to accelerate the phasing out of fossil fuels, such as the increased use of biomaterial and installations of wind turbines or hydropower infrastructure, also have implications for the land and water use of the Sámi people [47]. Due to these potential negative impacts of climate change and climate transition measures on the Sámi's traditional way of life, it is important that their right to be consulted is always respected. In terms of implementing the TJTP for Norrbotten, public authorities have begun the process of presenting and discussing proposals in meetings with the Council of the European Social Fund (ESF Council), Vinnova, and the Geological Survey of Sweden (SGU), and the Sámi Parliament. Dialogue with the Sámi Parliament is particularly significant as climate change has a major impact on the conditions for Sámi culture and land use. Nonetheless, Sámi representatives have argued that policies and legislation to support the climate transition have so far not adequately reflected their interests and that their traditional rights have not been respected. It is increasingly considered that, in order to increase the legitimacy of the climate transition process in Norrbotten, the Sámi community needs to be more strongly involved in decision making processes in the region, particularly in those related to land management issues [48].

*6.3. Governance Analysis*

Sweden has a decentralized system of governance in which regional and local public authorities play a central role in delivering public policies and services due to their close proximity to citizens. The institutional stakeholders at regional and local level in the three counties include the County Administrative Boards (CAB)—a national government authority operating in 21 counties with a responsibility for coordinating the climate transition work at regional and local level and ensuring that decisions from parliament and the Government are implemented in the counties. The CABs are also tasked with coordinating the work on regional climate and energy strategies, but the responsibility is shared. Regions and municipalities (self-governing local authorities) have a central role in the climate transition, not least in relation to strategic and physical planning, regional development work, education, stakeholder involvement, and advisory aspects. There are also local climate and environmental action plans at municipal level with localized climate policy targets [49]. Local authorities also play an important role in facilitating stakeholder coordination and promoting the conditions for collaboration between actors in the region, such as industries, businesses, associations, and other NGOs in relation to regional development. Local authorities in Gotland, Norrbotten, and Västra Götaland have facilitated dialogue between these key stakeholders in the development of regional and climate energy strategies [7].

Joint efforts between public authorities, regional and local companies, professional associations, and industry-relevant players are required to ensure that the climate transition takes place while maintaining the competitiveness and value of regions [31]. In all three Swedish just transition regions analyzed, high levels of cooperation and social capital have been built up through various EU and national innovation projects. In Norrbotten, the Hybrit Initiative, Reemap Project, and Sustainable Underground Mining Project have brought together key industries including SSAB, LKAB, and Vattenfall to develop carbon free production processes and circular economy initiatives. In Gotland, Cementa AB engages in various partnerships and R&D initiatives for the transition to fossil-free industrial processes. The industrial climate transition initiative Fossil-Free Sweden has set out a roadmap for a climate neutral cement industry in Sweden. The initiative has been led by Cementa AB [30]. The roadmap also notes that the construction sector and mining industries are closely linked to the cement industry's transition. Furthermore, Cementa AB, SMA Mineral AB, and Vattenfall, among others, collaborate under the umbrella initiative CemZero, conducting pilot studies and investigating the preconditions for climate neutral production processes. CemZero involves three research projects carried out in collaboration across the public and private sector as well as academia (Umeå University), partly funded by the Swedish Energy Agency [50]. The business collaboration Tillväxt Gotland also enables industrial actors to collaborate via the Industrigruppen Gotland in matters related to business growth, skills, and competence supply issues and transition.

In Västra Götaland, several science parks gather research and academia, public actors, businesses, and NGOs for the energy and climate transition. For example, Johanneberg Science Park hosts the West Swedish Chemical and Materials Cluster. High levels of cooperation and social capital are demonstrated through numerous collaboration projects in the region. The long-term project Climate Leading Process Industry supports the transition in a region with intense production of chemicals and materials. Here, the large chemical companies in Stenungsund (Adesso Bioproducts, Borealis, INOVYN, Nouryon, Perstorp) and Västra Götalandsregionen work together with a joint vision on Sustainable Chemistry by 2030. The West Sweden Chemicals and Materials Cluster includes research and innovation actors besides the chemical industry companies themselves, such as Chalmers University of Technology, SP Technical Research Institute of Sweden, as well as companies from other industry sectors, such as Renova and Göteborg Energi.

In Sweden, stable energy systems and electricity networks are key factors for the transition. For instance, it is estimated that the new HYBRIT plant planned in Gällivare and the H2 Green Steel plant in Boden together can increase the yearly electricity consumption

in Sweden by around 45–50 percent: this will require significant expansion of the renewable energy production [51]. This process is parallel to similar developments in the other Swedish regions. In Västra Götaland, the capacity of the electricity networks risks being insufficient in the short term [7]. In Gotland, developing the infrastructure that supports the energy transition and electrification of the cement production is also a key challenge. Cementa's need for electricity is expected to increase tenfold by 2030, as a result of the transition to carbon dioxide neutrality [7]. Improved transmission capacity of electricity between Gotland and the mainland is a prerequisite for large-scale electrification of the industry. A new cable connection needs to be in place within the next 10 years to align with Cementa's transition timeline. The relatively short time frame is emphasized as a key challenge in terms of planning, permits, and establishment. The design of the cable and investments is being investigated by Svenska Kraftnät, Vattenfall, Eldistribution, and Gotland Energy (GEAB).

Public authorities in all three regions work closely with higher education institutes. Gothenburg University, Chalmers University of Technology, Luleå Univeristy of Technology, and Uppsala University Campus Gotland carry out multidisciplinary research projects and have provided knowledge and expertise in the drafting of the regional energy and climate strategies. Municipalities, lower education providers, companies, and universities also play an important role in ensuring that there is a good match between access to a skilled workforce and employers' demand for skills. Public authorities in Gotland work with Uppsala University Campus Gotland and Teknik College Gotland to ensure that higher education and vocational training programs meet the needs of local industries and businesses. Norrbotten's Regional Competence Council coordinates actions on skills development and governance in the region. The task force is made up of municipalities, Swedish public employment service, Region Norrbotten (the County Council), the County Administrative Board, Luleå University of Technology (LTU), and vocational education schools. LTU also has an extensive range of training aimed at the steel industry's value chain and gathers research in mining and process technology. The Mining and Steel Industry Research Institute, Swerim, with process engineering and equipment in Luleå, is an important link between academia and industry. In Västra Götaland, there is a strong foundation for municipal collaboration projects to ensure skills supply in different sectors and industries in the region. These include for example Validation West (Validering Väst) that supports structure in the region to strengthen the skills validation process. Throughout the municipal associations branch, specific Competence Councils are also set up as a collaboration arena, ensuring coordinated efforts between labor market actors, academy, sectoral organizations, and other regional actors. The Gothenburg Region also has a competence hub to ensure coordinated efforts to support individuals and business life in transition.

## 7. Discussion

The core findings from the analyses presented above have informed and guided the preparation of the EU TJTPs for the Swedish transition regions of Gotland, Norrbotten, and Västra Götaland. These regional assessments highlight that the just transition has an important spatial dimension as each Swedish region is different in terms size, governance structure, and socio-economic dynamics. Gotland is a small rural and tourism-dependent economy that has very specific economic environment conditions determined by its insularity. Norrbotten is an extractive and increasingly technology and innovation-driven economy that is performing rather well and will probably keep doing so in a global context characterized by a transition from oil-based to metal-based energy carriers. Västra Götaland is a leading economy in the Nordics, with a broad industrial basis and a long-lasting tradition in innovation and technology-oriented productions. The diverse nature of Swedish regions makes it important to note that there is no one-size-fits-all model for just transition planning. This is largely reflected in the TJTPs that have been tailored to meet regional and territorial specificities and needs requirements.

Trade unions and civil society groups have voiced concern that the Swedish just transition is too focused on the economic and technical dimensions of climate policies, while neglecting the potential regional social impacts of transition. An assessment of the main transition actions outlined within the draft regional TJTPs plans suggest that this criticism is not unjustified, as they largely focus on developing the technical infrastructure required for reducing emissions in fossil fuel-dependent industries. The economic and technical imperative at the heart of the regional action plans has been driven by the national government during the TJTP development process.

During TJTP development discussions, this proved a source of tension with the EU who were more inclined to consider the social elements of the transition. The national government relented on its position and accepted the introduction of more social-orientated actions into the TJTPs when the regional territorial analysis highlighted that reskilling and retraining was essential in all Swedish transition regions. While some more socially targeted actions have been introduced within the plans, most of the actions remain focused on the technological enablers of the transition. This confirms the thesis that, also in Sweden, the national government tends to dominate multi-level discussions related to EU Cohesion Policy funding, with sub-national regional and local actors being constrained by political asymmetries of power [21,52].

Even though no job losses are anticipated within key regional industries, our analyses have highlighted some important social development trends within each region that might impact on the effective implementation of just transition plans. These issues need to be more clearly addressed within the regional TJTPs, including recommendations and solutions for overcoming these obstacles. Specific regions are more vulnerable to the effects of transition not only because of the regions' dependence on industries with high emissions, but also because of intrinsic conditions. Regions outside larger cities are likely to be more affected by industrial decline because of social development stressors such as the shrinking population caused by low birth rates and outmigration to urban areas, which make them more vulnerable to external shocks and policy reforms such as those brought about by the climate transition. Similarly, low-skilled workers are usually more vulnerable since it is more difficult to adapt to other occupations and because many affected industries are located in regions where the economy is not diversified.

Indeed, ensuring that companies in the cement and limestone, petro-chemical, and steel industries have access to the right skills and competencies is of fundamental importance for green just transition in Sweden. The TJTPs outline specific actions for skills development, but they are vague. Regional and local policy makers and stakeholders in Gotland, Västra Götaland, and Norrbotten can add further specification by focusing on four broad strategies to ensure future competence supply for the green just transition. Possible actions include: (1) providing access to higher education, re-training, and upskilling courses expanded and offered to all population groups; (2) encouraging people who are currently inactive or unemployed to upgrade their skills, facilitating their integration into the labor market; (3) developing tailored programs to attract skilled staff from other parts of Sweden and from abroad to fill competence gaps within regional labor markets; and (4) in addition to the already existing industry road maps, social road maps could be created to highlight the social dimension of the transition and to form collective social action.

The key governance structures and stakeholders needed for the development and implementation of climate transition policies in Sweden are highlighted within the TJTPs. This can be considered a key enabler for the technical solutions. For instance, in the preparation of the new national electrification strategy it has been emphasized that ensuring the electrification process to support the climate transition requires improved collaboration between governance levels as well as streamlining and expansion of the network [53]. However, the plans do not provide any detailed specification on the role and responsibilities of these governance levels and actors at different stages of the transition policy cycle. According to Swedish climate policies, climate transition processes should be open and inclusive of all stakeholders and citizens, particularly underrepresented minority groups [5].

There is little information within the TJTPs regarding public consultation on the transition proposals. There is, therefore, a need for further analysis on how best to include the public in the transition process from both a policy input and policy dissemination perspective. This is particularly the case in the Norrbotten region where there are still tensions with the indigenous Sámi community regarding transition proposals and discussions with the Sámi Parliament remain ongoing. In order to meet the goal of 'leaving no one behind' during the climate transition, but also to avoid breaching the legal rights and prerogatives of the Sámi people, their involvement should be enhanced.

Our territorial analysis reveals that each region has the governance structures and stakeholders needed to develop and implement transition policies. Existing governance structures are based on strong links between national government, national agencies and regional public authorities, and high levels of social capital and collaboration between key stakeholders, including industries, businesses, high education institutions, labor unions, and civil society organizations. Governance challenges remain, however, including a lack of dialogue and co-operation between the regional and local electricity grid-owners regarding conditions for energy supply in the comprehensive or physical planning that the municipalities carry out. As the structures for coordination are weak, there is a risk that electricity users, community planners, and network companies will not be made aware of the limitations that exist in the electricity network. Lengthy permitting procedures may also affect many sectors concerned by climate transitions. Current procedures have been designed to ensure compliance with highly conservative technical and safety regulations in sectors such as energy production, storage, and distribution. Permitting procedures should be simplified in a way that safety and stability are maintained while ensuring that the processes are significantly shortened. This example further reinforces the idea that the Swedish just transition process should pay more attention to the governance dimension of transitions to also enable its technical components. Further cross-sectoral collaboration and communication will be required to overcome these challenges.

## 8. Conclusions

With the introduction of the EU's Green Deal and Just Transition Mechanism, the concept of the 'just transition' is starting to grow and gain momentum. There are many lessons to be drawn for both Sweden and other EU member states from the Swedish experience of developing TJTPs. Most significantly, it is important that EU action plans, and other climate and energy related policies, are developed based on a well-defined just transition concept. Because of the broad nature of the just transition notion, it is used and interpreted differently in different contexts. It is important to balance between acknowledging the rights of the EU, nation states, and regions to interpret and implement the concept in different ways, while also attempting to provide some coherence, so that plans can be implemented smoothly in a way that meets the needs of local citizens.

Swedish policymakers and the TJTPs they are developing need to fully recognize the different technical, social, and spatial elements of the shift towards climate neutrality. In the Swedish case, there has been a tendency for national policymakers to prioritize the more technical elements of the transition, driven by the economic cost of helping industries transfer to climate neutral, carbon-free technologies. It is not surprising that the actions outlined in the TJTPs emphasize the technical and economic imperatives of transition as the TJTP development process was driven by representatives of the Swedish national government, regional public authorities, and sectoral actors. Indeed, the actions outlined in the TJTPs are entirely consistent with their climate policies and roadmaps which are focused on the technical aspects of shifting to carbon neutral technologies and processes and neglect the potential social impacts of the transition. This position was exacerbated by the processes used in the formulation of TJTPs, as societal groups, NGOs, and citizens were largely excluded from this process, with the notable exception of the Sámi Parliament in Norrbotten. The European Commission note that the transition can be successful only if policies are designed with the involvement of citizens and accepted by them [2]. More

specifically, an 'active social dialogue' is recognized as an essential element to ensure that the transition is successful and accepted by workers and companies [2]. In Sweden, greater consultation of societal groups and citizens in the TJTP process might have acted as a counterweight to the political and technical priorities of the Swedish government, regional authorities, and sectors, creating a balance between the technical and social actions outlined in the TJTPs. It is, therefore, vital that social groups and local citizens be consulted in any evaluation and amendments of the plans if they are to be fully accepted by society.

Transition plans need to strike a balance between these important economic and technical imperatives, while ensuring that the social justice and spatial dimensions of the transition are not ignored. To make sure that the social and spatial elements are not neglected, policymakers and practitioners could, one, develop social roadmaps that highlight the social challenges posed by climate policies within each transition region, and two, ensure that climate transition policies are based on areas of regional resilience strengths and opportunities. These measures would make policies responsive to local needs and objectives, and more accessible to citizens in helping overcome the challenges posed by regional socio-economic trends, including an aging work force, lower education levels, outmigration, and gender imbalances.

The social impacts of the transition should be minimal in the Swedish transition regions if, as noted by interviewees, there are no resultant job losses in the industries most affected by the shift to carbon neutral technologies. Furthermore, the Swedish welfare system is considered to be well equipped to deal with the possibility of job losses and the resultant social challenges presented by increasing unemployment. In this regard, the strength of the Swedish welfare system has played a significant role in driving the political decision of the Swedish national government to focus TJTP actions on the more technical and economic aspects of transition, including covering the costs of the development and implementation of carbon neutral technologies and processes in key industries. The structural limitations of the Just Transition Fund identified by Sabato and Fronteddu [19] have not played such a significant role in influencing the content of the TJTPs in Sweden. The narrow focus of the Just Transition Funds on skills and retraining they outline is potentially beneficial to Swedish transition regions and industries, but has largely been a secondary consideration within the actions outlined in the TJTPs, despite regional impact assessments highlighting the need for both youth and elderly reskilling within Swedish transition regions. In countries with less generous welfare systems, the narrow focus on the Just Transition Funds on skills and retraining will not help overcome the more short-term challenges presented by unemployment. There is, therefore, a need to ensure that the focus on the Just Transition Funds on skills and retraining is not viewed as an alternative to traditional social welfare measures and that the Just Transition Funds effectively complement and fill gaps in national and regional level welfare policies [19].

An effective climate transition process requires smooth collaboration across multiple levels of governance. Ensuring that EU, national, and regional level goals and objectives are represented within transition plans requires decision-making based on equality and reciprocal persuasion in which actors try to understand each other's incentives and underlying assumptions. Strong regional and local leadership is particularly important for facilitating open and inclusive dialogue between key regional and local levels stakeholders and citizens. The Swedish case regions have shown that effective collaboration is often based on the social capital built up through existing regional networks, mainly involving industries, businesses, and higher education institutes. However, it is also vital that transition processes are open and transparent to facilitate participatory governance throughout the policy cycle which involves labor unions, civil society organizations, and minority groups. Moving forward, the development of national and regional climate transition platforms might be one way to maintain smooth and effective collaborations across governance levels. Such platforms could help establish sectoral synergies and business value chains, coordinate public and private financing, identify existing and newly emerging socio-economic challenges and opportunities, and monitor the impacts of the transition. Collaboration in

relation to these issues will be essential if the technical, social, and territorial dimensions of the just transition are to be addressed in a balanced and equal manner.

An important criterion for just transitions is that 'no one is left behind', but the transition plans should also ensure that environmental burdens and social impacts are not transferred to other regions. The Swedish examples presented here show how that, even if the socio-economic impacts of the transitions might be large in some settings, in most cases, these impacts are highly localized and could be totally or partially overcome by the strong economic inertia and innovation capacity of the Swedish economy. In some cases, it might be even easier to replace jobs in certain activities by occupations in other sectors with smaller climate footprints rather than investing in the transformation of the problematic energy-intensive industries. Still, the products provided by these industries, including cement, steel, and chemical compounds, will continue to be demanded and consumed by the Swedish economy. Delocalizing the production of these materials to other regions would not reduce the global environmental burden and would not contribute to tackling the climate emergency either. In this sense, the just transition concept acquires an ethical dimension that goes beyond the social justice principle that lays at its core. It is essential that the just transition plans in Sweden and elsewhere in Europe take up this challenge and contribute to ensuring that the climate transitions are performed in situ, without alienating the employment basis, production capacity, and sectoral specialization of the affected regions.

**Author Contributions:** Conceptualization: all authors; methodology: all authors; software: not applicable; validation: all authors; formal analysis: all authors; investigation: all authors; resources: not applicable; data curation: all authors; writing—original draft preparation: all authors; writing—review and editing: all authors; visualization: not applicable; supervision: not applicable; project administration: not applicable; funding acquisition: not applicable. All authors have read and agreed to the published version of the manuscript.

**Funding:** This research received no external funding.

**Institutional Review Board Statement:** Not applicable.

**Informed Consent Statement:** Not applicable.

**Data Availability Statement:** Not applicable.

**Acknowledgments:** We would like to thank the journal editors and three reviewers for their helpful and constructive feedback that helped to improve the structure and focus of the paper immeasurably. We would also like to thank our colleagues who helped in the preparation of the Swedish Territorial Just Transition Plan drafts, particularly those at Tillväxtverket, Trinomics, and regional authorities of Gotland, Norrbotten and Västra Götaland.

**Conflicts of Interest:** The authors declare no conflict of interest.

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
