# Peer review of "Towards a Territorially Just Climate Transition—Assessing the Swedish EU Territorial Just Transition Plan Development Process"

_sustainability, doi:10.3390/su13137505_

Round 1
Reviewer 1 Report
The paper characterises the analysis as „ethnographic” which I believe is not the case, as the analysis is mainly reliant on process tracing and institutionalist analysis. The authors should revise their methodological description and detail the approach.
The paper mentions that three out of four TJTP eligible regions in Sweden were covered by the analysis, but does not clearly explains the criteria of selection.
The reference list should have a homogenous styling.
Author Response
Reviewer 1 Comment 1
The paper characterises the analysis as „ethnographic” which I believe is not the case, as the analysis is mainly reliant on process tracing and institutionalist analysis. The authors should revise their methodological description and detail the approach.
Authors Response
We thank reviewer one for the comment. We have addressed this in the abstract, introduction and main methodology section in the article by making it clear, as reviewer one suggests, that the analysis is primarily driven by the socio-economic and institutional impact assessment data which is supported by process observations and interviews. In the methodology section, we also have specified in more detail which research methods and data were used in relation to each part of the impact assessment.
Reviewer 1 Comment 2
The paper mentions that three out of four TJTP eligible regions in Sweden were covered by the analysis, but does not clearly explains the criteria of selection.
Authors Response
We have now inserted a line in the introduction which states clearly why only three of the four Swedish transition regions are covered by the analysis. The reason for this is that the Swedish government had already conducted their own analysis in the Västerbotten region, so an additional impact assessment was not required from the research team in this area.
Reviewer 1 Comment 3
The reference list should have a homogenous styling.
Authors Response
We have changed the endnotes from the roman numerals used in the layouted version we received to numbers. We have also been through the journal referencing guide to ensure that the references are aligned with these requirements.
Reviewer 2 Report
This article is relevant and original, in particular for two reasons. First, it deals with the notion of just transition, a notion that – as stated by the authors – is gaining an increasing relevance in political and policy debates, especially in the European Union. Second, it engages in an empirical analysis of the implementation of one of the main instruments created at the EU level in order to promote just transition in specific territories of the Union: the Just Transition Mechanism and its Just Transition Fund. The authors investigate the process of elaboration and the contents of the Territorial Just Transition Plans of three Swedish regions, thus performing what is – in my knowledge - one of the first empirical investigations of the actual implementation on the ground of the Just Transition Mechanism and of the Just Transition Fund. In this sense, the article has the potential to make a contribution to both the academic literature on just transition (including the literature on EU policies) and to policy-makers engaged in elaborating territorial just transition strategies.
This said, before a possible acceptance for publication, a number of important points should be better clarified and investigated more in-depth by the authors. These concern: i) the analytical framework – based on the identification of three conceptual dimensions of just transition – and the use of the notion of just transition in key EU documents; ii) methodological aspects and empirical analysis; iii) the interpretation of the findings and the conclusion.
Analytical framework and just transition in key EU documents
The authors rightly note that just transition is ‘an amorphous concept’ and, in an attempt to pursue conceptual clarity and to operationalise the concept for the empirical analysis, they identify three dimensions of the notion of just transition: technical, social justice, and spatial dimension. While I find these three dimensions appropriate and useful, I would also invite the authors to reflect further on the social justice dimension and on its link with the territorial dimension. This would be, in my opinion, particularly important in Section 3 on ‘The EU and Just Transition’. When it comes to the social justice dimension of just transition, the authors distinguish between a procedural justice element (process) and a distributional justice element (simplifying: ensuring that both the opportunities and burdens of the transition are fairly shared among social groups and territories). On how to achieve the latter objective, however, some observers have pointed at some tensions and potential contradiction characterising EU policy discourses and initiatives on just transition, i.e. tensions between the need to implement social investment policies (e.g., active labour market policies, training and re-skilling policies to increase workers’ employability in a greener economy) and the need to ensure the protection of citizens through traditional social protection policies (e.g., unemployment and minimum income benefits) and to ensure quality employment. In this perspective, the understanding of just transition in the EGD would be unbalanced towards a social-investment-oriented and territorially focused approach, while the link with a more general framework such as the European Pillar of Social Rights would risk to be quite weak (cf., for instance, Sabato and Fronteddu (2020), ‘A socially just transition through the European Green Deal?’, ETUI Working Paper 2020.08, Brussels: European Trade Union Institute) . Besides EU policies, these possible tensions have been highlighted by the International Labour Organisation in their 2013 Resolution (quoted by the authors) and in their 2015 Guidelines on Just Transition. In particular, the latter document highlights the need to ensure coherence between various policy domains, avoiding the risk to reduce the transition to a matter of skill development and retraining initiatives, and to carefully consider the interaction of policies developed at different territorial levels (e.g., local transition policies and broader welfare policies). These aspects will be increasingly important in EU policy-making, since national Just Transition Plans are expected to be integrated in and coherent with national reforms presented by the Member States in their National Recovery and Resilience Plans.
Consequently, in my opinion, the authors could consider engaging more, both at the theoretical level and in the analysis of the EU framework in Section 3, in problematising these distinctions and potential tensions pertaining to the social justice component of just transition and its relationship with the territorial component.
Methodology and empirical analysis
I think that the methodology used (mixed qualitative and quantitative approach) is quite solid. However, I should admit that, after having read carefully the paper a few times, it has not been immediately clear to me what is the main source of the empirical analysis in Section 6 (so, to what that methodology has been actually applied). In my understanding, Nordregio was involved in the elaboration of the TJTPs of the three Swedish regions, supporting the other actors in the elaboration of those plans. In this context, Nordregio conducted its analysis (based on qualitative and quantitative methods), with a view to inform the preparation of the TJTPs. At this point, however, things become less clear to me: is the empirical Section 6 based on the Nordregio’s report to Swedish policy-makers or on the actual analysis of the resulting TJTPs drafted by public authorities (or on a mix of the two)? It is crucial that the authors clarify this aspect. Indeed, as it stands now, Section 6 may result quite confusing: it containing a number of (interesting) factual information, opinions and assessments but the reader does not fully understand whether all that emerges from the authors’ previous analyses that contributed to the TJTPs, from the draft TJTPs itself or from the authors’ analysis of the latter. Additionally, once clarified this aspect the authors might consider moving what in Section 6 is their assessments and opinions to the Discussion in Section 7.
Interpretation and conclusions
In this article, relying on an ethnographic observation of the process of elaboration of the TJTPs of the three regions, the authors aim at establishing ‘whether the content and actions outlined in the TJTPs were driven by the technical, social or spatial dimensions of a just transition’. In this respect, they conclude that technical considerations are prevalent in the TJTPs, the territorial dimension has been taken into consideration to a good extent (no ‘one-size-fits-all’ approach), while important limitations do emerge when looking at the social justice dimension of the TJTPs. On the basis of these findings, the authors also draw some ‘lessons’ (page 18) for Swedish and EU policy-makers. I think that the findings are quite interesting and they could help in identifying lessons and providing recommendations for policy-makers. However, in order to do so, I would invite the authors to reflect more and to be more explicit on the elements (both positive and negative) of the process that, in their opinion, have led to the outcomes identified. For instance, the authors should answer more explicitly questions such as: have those outcomes been the result of the type of actors involved in the process? How much technical and political considerations have affected the final outcome ? To what extent the outcomes have been determined by factors outside the regional/national process of elaboration of the TJTPs such as requirements already present in the EU regulation on the Just Transition Fund (incl. possible structural limitations of the policy instrument)?
In addition to the more general considerations above, please find some more specific aspects to be addressed:
- As the authors specify in the article, the three TJTPs are now in the process of being reviewed by the Swedish Government and the EU. So, the authors should probably refer to them (at least in the Introduction) as ‘draft TJTPs’.
- I would suggest that the authors add some detail on how and why those three regions were selected for support from the Just Transition Fund. This would add some context useful for the reader.
- Sometimes (lines 55, 69, and 427), the authors write ‘Just Transition Funds’ (plural). I think it should be ‘Just Transition Fund’ (singular).
- Page 2/lines 80-83: ‘Nordregio’s research team was established by DG Reform to directly assist and support national and regional actors in the preparation of the Swedish TJTPs’. Maybe using the verb ‘establishing’ is a bit confusing. Do the authors mean that Nordregion’s research team was ‘tasked’ by DG Reform (or a verb of an equivalent meaning)?
- Please check page 9 lines 374-378 (same sentence repeated twice); - page 10 Title of Table 2 (possible typo); page 13 lines 535 – 536 (the sentence ‘has fallen from 32,000 persons in 2009 to around 32,700 persons in 2019’ does not seem coherent. I would say that it is an increase, not a fall).
Author Response
Reviewer 3 Comments 1
The authors rightly note that just transition is ‘an amorphous concept’ and, in an attempt to pursue conceptual clarity and to operationalise the concept for the empirical analysis, they identify three dimensions of the notion of just transition: technical, social justice, and spatial dimension. While I find these three dimensions appropriate and useful, I would also invite the authors to reflect further on the social justice dimension and on its link with the territorial dimension. This would be, in my opinion, particularly important in Section 3 on ‘The EU and Just Transition’. When it comes to the social justice dimension of just transition, the authors distinguish between a procedural justice element (process) and a distributional justice element (simplifying: ensuring that both the opportunities and burdens of the transition are fairly shared among social groups and territories). On how to achieve the latter objective, however, some observers have pointed at some tensions and potential contradiction characterising EU policy discourses and initiatives on just transition, i.e. tensions between the need to implement social investment policies (e.g., active labour market policies, training and re-skilling policies to increase workers’ employability in a greener economy) and the need to ensure the protection of citizens through traditional social protection policies (e.g., unemployment and minimum income benefits) and to ensure quality employment. In this perspective, the understanding of just transition in the EGD would be unbalanced towards a social-investment-oriented and territorially focused approach, while the link with a more general framework such as the European Pillar of Social Rights would risk to be quite weak (cf., for instance, Sabato and Fronteddu (2020), ‘A socially just transition through the European Green Deal?’, ETUI Working Paper 2020.08, Brussels: European Trade Union Institute) . Besides EU policies, these possible tensions have been highlighted by the International Labour Organisation in their 2013 Resolution (quoted by the authors) and in their 2015 Guidelines on Just Transition. In particular, the latter document highlights the need to ensure coherence between various policy domains, avoiding the risk to reduce the transition to a matter of skill development and retraining initiatives, and to carefully consider the interaction of policies developed at different territorial levels (e.g., local transition policies and broader welfare policies). These aspects will be increasingly important in EU policy-making, since national Just Transition Plans are expected to be integrated in and coherent with national reforms presented by the Member States in their National Recovery and Resilience Plans.
Authors Response
We thank the reviewer for pointing out the tensions between the different dimensions of a just transition, which was extremely helpful, as was the references that they suggested we read and include. We have addressed this issue in the conceptual section and EU analysis section as requested by the reviewer by acknowledging some contradictions between the ILO guidelines and the EU policy approach in relation to the focus on skills and training over traditional unemployment welfare assistance.
Reviewer 3 Comment 2
I think that the methodology used (mixed qualitative and quantitative approach) is quite solid. However, I should admit that, after having read carefully the paper a few times, it has not been immediately clear to me what is the main source of the empirical analysis in Section 6 (so, to what that methodology has been actually applied). In my understanding, Nordregio was involved in the elaboration of the TJTPs of the three Swedish regions, supporting the other actors in the elaboration of those plans. In this context, Nordregio conducted its analysis (based on qualitative and quantitative methods), with a view to inform the preparation of the TJTPs. At this point, however, things become less clear to me: is the empirical Section 6 based on the Nordregio’s report to Swedish policy-makers or on the actual analysis of the resulting TJTPs drafted by public authorities (or on a mix of the two)? It is crucial that the authors clarify this aspect. Indeed, as it stands now, Section 6 may result quite confusing: it containing a number of (interesting) factual information, opinions and assessments but the reader does not fully understand whether all that emerges from the authors’ previous analyses that contributed to the TJTPs, from the draft TJTPs itself or from the authors’ analysis of the latter. Additionally, once clarified this aspect the authors might consider moving what in Section 6 is their assessments and opinions to the Discussion in Section 7.
Authors Response
Many thanks for the comment. The origin of our empirical work was in fact not clearly described in our paper. We have addressed the issue by expanding the introduction to the Research methods section (S4), which now provides further details on each analytical component underpinning our research. Last but not least, we have revised the text in Section 6 trying to avoid personal judgement and opinions. AS suggested, all potentially subjective claims have been either removed or moved to the Discussion section.
Reviewer 3 Comment 3
In this article, relying on an ethnographic observation of the process of elaboration of the TJTPs of the three regions, the authors aim at establishing ‘whether the content and actions outlined in the TJTPs were driven by the technical, social or spatial dimensions of a just transition’. In this respect, they conclude that technical considerations are prevalent in the TJTPs, the territorial dimension has been taken into consideration to a good extent (no ‘one-size-fits-all’ approach), while important limitations do emerge when looking at the social justice dimension of the TJTPs. On the basis of these findings, the authors also draw some ‘lessons’ (page 18) for Swedish and EU policy-makers. I think that the findings are quite interesting and they could help in identifying lessons and providing recommendations for policy-makers. However, in order to do so, I would invite the authors to reflect more and to be more explicit on the elements (both positive and negative) of the process that, in their opinion, have led to the outcomes identified. For instance, the authors should answer more explicitly questions such as: have those outcomes been the result of the type of actors involved in the process? How much technical and political considerations have affected the final outcome ? To what extent the outcomes have been determined by factors outside the regional/national process of elaboration of the TJTPs such as requirements already present in the EU regulation on the Just Transition Fund (incl. possible structural limitations of the policy instrument)?
Authors Response
In response to these comments, we have brought two new paragraphs into the conclusion (in red font) which address the question posed by the reviewer. The first highlights weaknesses with the jut transition plan process particular the limited role of societal groups and citizens in the formulation process and the potential impact on policy acceptance. The second paragraph assesses whether the structural limitations of the just transition fund impacted on the process. We argue that in the Swedish process this has not been the case, but the structural limitations the reviewer outlines could be more significant in EU members states without such generous national/regional level welfare systems.
Reviewers Comments 4
- As the authors specify in the article, the three TJTPs are now in the process of being reviewed by the Swedish Government and the EU. So, the authors should probably refer to them (at least in the Introduction) as ‘draft TJTPs’ – corrected in the text
- I would suggest that the authors add some detail on how and why those three regions were selected for support from the Just Transition Fund. This would add some context useful for the reader – sentence added in the introduction to address this point.
- Sometimes (lines 55, 69, and 427), the authors write ‘Just Transition Funds’ (plural). I think it should be ‘Just Transition Fund’ (singular) – corrected in the text.
- Page 2/lines 80-83: ‘Nordregio’s research team was established by DG Reform to directly assist and support national and regional actors in the preparation of the Swedish TJTPs’. Maybe using the verb ‘establishing’ is a bit confusing. Do the authors mean that Nordregion’s research team was ‘tasked’ by DG Reform (or a verb of an equivalent meaning)? – corrected in the text
- Please check page 9 lines 374-378 (same sentence repeated twice); - page 10 Title of Table 2 (possible typo); page 13 lines 535 – 536 (the sentence ‘has fallen from 32,000 persons in 2009 to around 32,700 persons in 2019’ does not seem coherent. I would say that it is an increase, not a fall) – corrected in the text
Reviewer 3 Report
The paper assess the processes undertaken during the development of EU Territorial Just Transition Plans for three Swedish regions of Gotland, Norrbotten and Västra Götaland showing that a balance between the technical, social and spatial elements of a just transition is needed if policies are going to meet the requirements of local and regional citizens and provide sustainable socio-economic growth and environmental protection, without risks of delocalising energy-intensive processes to other regions. The subject is worth investigating and the paper can be considered for publication after a minor revision addressing the following issues.
Line 79: “in” must be corrected.
Lin 140: “seminal” must be corrected.
Line 352: Please consider to use „The Table 1 provides” instead of „The table one below provides”.
Line 354: please explain what means „Riksdag„
Table 1: please check if „healthcare and care” is correct.
Line405: Please consider to use „… in Table 2.” instead of „… in table two below”.
Line 464: please consider to delete „below”.
Lines 535-536: please check: “has fallen from 32,000 … to 32,700”.
Author Response
Reviewer 2 Comments
Line 79: “in” must be corrected – we have corrected this in the article
Lin 140: “seminal” must be corrected – we have corrected this in the article
Line 352: Please consider to use „The Table 1 provides” instead of „The table one below provides” – we have corrected this in the article
Line 354: please explain what means „Riksdag„ – we have corrected this in the article
Table 1: please check if „healthcare and care” is correct – we have corrected this in the article
Line405: Please consider to use „… in Table 2.” instead of „… in table two below” – we have corrected this in the article
Line 464: please consider to delete „below” – we have corrected this in the article
Lines 535-536: please check: “has fallen from 32,000 … to 32,700” – we have corrected this in the article